# Towards Sustainable and Dynamic Modeling Analysis in Korean Pine Nuts: An Online Learning Approach with NIRS

**DOI:** 10.3390/foods13172857

**Published:** 2024-09-09

**Authors:** Hongbo Li, Dapeng Jiang, Wanjing Dong, Jin Cheng, Xihai Zhang

**Affiliations:** 1College of Electrical and Information, Northeast Agricultural University, 59 Changjiang Rd., Harbin 150030, China; lihongbo@neau.edu.cn (H.L.); s221401016@neau.edu.cn (J.C.); 2College of Computer and Control Engineering, Northeast Forestry University, 26 Hexing Rd., Harbin 150040, China; jiangdapeng1992@nefu.edu.cn; 3College of Economics and Management, Northeast Forestry University, 26 Hexing Rd., Harbin 150040, China; dwj@nefu.edu.cn; 4National Key Laboratory of Smart Farm Technology and System, Northeast Agricultural University, 59 Changjiang Rd., Harbin 150030, China

**Keywords:** Korean pine nut, near-infrared spectroscopy technology, online learning, fat content detection, online multiplicative scatter correction, recursive partial least squares, uninformative variable elimination

## Abstract

Due to its advantages such as speed and noninvasive nature, near-infrared spectroscopy (NIRS) technology has been widely used in detecting the nutritional content of nut food. This study aims to address the problem of offline quantitative analysis models producing unsatisfactory results for different batches of samples due to complex and unquantifiable factors such as storage conditions and origin differences of Korean pine nuts. Based on the offline model, an online learning model was proposed using recursive partial least squares (RPLS) regression with online multiplicative scatter correction (OMSC) preprocessing. This approach enables online updates of the original detection model using a small amount of sample data, thereby improving its generalization ability. The OMSC algorithm reduces the prediction error caused by the inability to perform effective scatter correction on the updated dataset. The uninformative variable elimination (UVE) algorithm appropriately increases the number of selected feature bands during the model updating process to expand the range of potentially relevant features. The final model is iteratively obtained by combining new sample feature data with RPLS. The results show that, after OMSC preprocessing, with the number of features increased to 100, the new online model’s $R^{2}$ value for the prediction set is 0.8945. The root mean square error of prediction (RMSEP) is 3.5964, significantly outperforming the offline model, which yields values of 0.4525 and 24.6543, respectively. This indicates that the online model has dynamic and sustainable characteristics that closely approximate practical detection, and it provides technical references and methodologies for the design and development of detection systems. It also offers an environmentally friendly tool for rapid on-site analysis for nut food regulatory agencies and production enterprises.

## 1. Introduction

Korean pine nut is the seed of *Pinus koraiensis* Sieb. et Zucc., which is resistant to cold and prefers slightly acidic or neutral soil. It is mainly produced in the Changbai Mountain area in northeast China, including Jilin and Xiaoxing’anling, with an altitude range of 150–1800 m, in forests with warm, cold, and humid climates. It is also distributed in Japan (Honshu), Korea, and Russia (Amur, Khabarovsk) [1]. Pine nuts are rich in unsaturated fatty acids beneficial for human health  [2], which makes them and other products such as pine nut oil popular among consumers.

The unsaturated fatty acids in Korean pine nuts are an important indicator of their nutritional value. As a result, Korean pine nuts occupy a high position among nut foods. From the perspective of consumption, the fat content is a direct indicator of the fatty acid content and the oil yield [2,3]. Therefore, the fat content can be used as the detection target. Traditional chemical detection methods are laborious and time-consuming, making them impossible to use for detection of large quantities, and improper treatment of waste liquid can pollute the environment. Rapid and nondestructive detection of pine nut fat content can help classify its edible grade. The most important indirect analysis characteristic of near-infrared spectroscopy (NIRS) is the regression detection of specific substance content in samples. The research on NIRS mainly focuses on ensuring the accuracy of the detection results as quickly as possible in real time. By reducing the number of steps in the detection process, simplifying the operation, avoiding the generation of a large amount of waste and harmful reagents, and making precision detection less dependent on strict experimental conditions, it is expected to achieve a high level of popularity, with wide sample coverage and low equipment and operation thresholds for manufacturers. This nondestructive detection technology is completely capable of detecting the fat content of Korean pine nuts [4].

Currently, there is limited research on the application of NIRS for analyzing Korean pine nuts. However, due to its advantages such as rapidity, environmental friendliness, and ease of operation, NIRS has found widespread use in analyzing agricultural and food products [4,5,6]. In recent years, researchers have increasingly utilized spectroscopic techniques to analyze the fat content in various foods, such as soybeans, meat, and dairy products [7,8,9,10]. NIRS can be used to analyze samples using diffuse reflectance spectroscopy. Spectroscopy combined with chemometrics has been extensively employed in testing nut quality, encompassing qualitative tasks such as variety identification and adulteration detection, as well as quantitative analysis of substance content. Existing research demonstrates the traceability of multiple varieties of walnuts from different production areas using NIRS [11]. In a study on peanuts and blocky nuts, NIRS successfully distinguished among peanuts, pine nuts, almonds, sesame seeds, and flax seeds [12]. Moreover, in the domain of substance content analysis, NIRS accurately detects higher levels of protein, water, and other substances in nuts and characterizes lower levels of water-soluble sugars and AFB1 [13,14,15,16,17]. Additionally, NIRS can quantify the unique crispy texture of nuts, corresponding to physical properties such as fracture force, hardness, and elasticity modulus [14].

The prediction and classification of nuts have relied on various methods, including statistical techniques such as multiple linear regression (MLR), partial least squares (PLS) regression, and the naive Bayes algorithm; chemometric techniques such as first and second derivative, multiplicative scatter correction (MSC), and standard normal variate (SNV) algorithms [18,19]; and machine learning techniques such as different types of kernel smoothing methods, boosting methods, and additive models [20,21,22]. These models typically operate in a batch learning or offline learning mode. Traditional batch-style machine learning methods, however, are plagued by several significant limitations: (1) they exhibit low efficiency in terms of time and space costs; and (2) they demonstrate poor scalability for large-scale applications because the models typically require retraining from scratch with new training data.

Online learning, a subfield of machine learning, differs from traditional batch-style machine learning in that it aims to incrementally learn from sequential data. Online learning algorithms are easy to understand and implement, typically built on theories with rigorous regret bounds [23], and the algorithm can immediately update the prediction model for new data. Therefore, in large-scale food inspection, when the test data are input to the model in a sequential manner and the detection target may drift or evolve over time, online learning algorithms are usually more efficient and scalable than offline learning algorithms.

Online learning includes unsupervised learning and supervised learning, with unsupervised learning mostly using methods such as kernel PCA [24], kernel ICA [25], and manifold learning [26]. However, the spectral band extraction techniques for NIRS, including uninformative variable elimination (UVE), Monte Carlo uninformative variable elimination (MC-UVE), the successive projections algorithm (SPA), and the competitive adaptive reweighted sampling (CARS) algorithm [27], do not currently have online optimization algorithms based on increment. To address this issue, we propose an online detection model based on RPLS. We utilize UVE to extract spectral bands and adjust model parameters to align with the requirements of online learning. Additionally, we design an improved online preprocessing method to calibrate raw spectra. The main contributions are as follows:To address the issue of independent scatter correction not allowing the addition of new samples outside the original modeling dataset, we propose an online multiplicative scatter correction (OMSC) preprocessing algorithm. Inspired by the reference spectrum in MSC, we design a dynamic reference spectrum that can change with variations in the detection samples, enabling online correction of the original spectra.To address the problem of constantly changing datasets during online detection, which can lead to the problem of constantly changing feature bands, we use UVE to extract the spectral feature bands and expand the number of bands in the feature subset. During the iterative update process of the model, we analyze the impact of parameter settings on the coverage range of the selected feature bands and verify the necessity of adjusting the number of features.To address the issue of detecting newly added pine nut samples without rebuilding the model and to solve the problem of the original regression model performing poorly on samples from different batches, we conducted research on the sustainable use of offline models and established an online detection model based on RPLS.

The rest of this article is structured as follows. In Section 2, we describe the experimental setup, sample preparation, and details of the proposed method. Section 3 presents extensive experiments conducted on the dataset prepared for this study to validate the effectiveness of the online learning model. In Section 4, we compare the prediction performance of the original offline model with that of the online model on new samples. Furthermore, we examine whether the online approach proposed in this study enhances the model’s generalization ability.

## 2. Materials and Methods

### 2.1. Preparation of Materials and Dataset Partitioning

In accordance with the research requirements, the samples needed for the experiment were all purchased from the main production areas of *Pinus koraiensis* in Northeast China. The sample preparation mainly included pine nut selection, shelling, and kernel separation. Based on the principles of random sampling, chemometrics, and machine learning modeling requirements, the final sample size was determined. After the pine cones had matured, 100 mature and well-preserved pine nuts were randomly sampled as one group, with each group weighing about 20 g. A total of 120 groups of samples were collected and placed in separate sealed bags, numbered from #1 to #120. These samples were used to establish the offline model. For the online learning model, 75 new samples were needed, which were purchased from different batches of pine nuts. Starting from the first purchase, a small batch of freshly picked pine nuts was purchased from farmers every 3 days, and 5 groups of new samples were made following the above experimental steps. A total of 75 groups of samples were collected, numbered from #1 to #75. The sampling process and the final prepared samples are shown in Figure 1. All samples were properly stored away from light, waiting for the next step of spectral collection and chemical experiments.

During the data acquisition phase of pine nut processing, we conducted spectral detection, comparison, and analysis of pine nuts with and without their skins. Figure 2a,b illustrates the spectral data for skinned and unshelled pine nuts, respectively. It is evident that the spectral trends and absorption peaks are largely consistent between the two sets. Considering the conclusions drawn from Figure 2a,b, along with the ability of near-infrared spectroscopy to penetrate materials up to 0.1 mm [28], we ultimately decided to use unshelled pine nuts with skins for detection to ensure the integrity of the samples.

It should be noted that quantitative analysis of the fat content requires coordination with chemical analysis. Chemical analysis requires a certain amount of the sample to undergo a series of reactions and extractions to obtain valid data. Considering the potential loss during chemical analysis and the feasibility of collecting spectral data, each sample was standardized to 20 g.

### 2.2. Spectral Data Collection and Chemical Experiments

The NIRQuest512 Near Infrared spectrometer from Ocean Optics was selected for spectral data acquisition due to its robustness, high signal-to-noise ratio, high resolution, and capability for acquiring high-dimensional spectral data. Its wavelength range spans from 900 nm to 1700 nm, encompassing the spectral information necessary for analyzing the chemical bonds of fat in pine nuts. To ensure accurate data collection, it is crucial to maintain close contact between the pine nut sample and the probe fixture to prevent light leakage.

The NIRQuest512 near-infrared spectrometer from Ocean Optics was selected for spectral data acquisition due to its robustness, high signal-to-noise ratio, high resolution, and ability to acquire high-dimensional spectral data. The spectrometer operates within a wavelength range of 900 nm to 1700 nm, which is ideal for analyzing the chemical bonds in the fats of pine nuts. The light source for the spectrometer was the HL-2000 Tungsten Halogen Light Source. To minimize light leakage and ensure accurate data collection, we maintained close contact between the pine nut sample and the reflection probe fixture. The fiber optic accessories included VIS-NIR fibers with core sizes of 200, 400, and 800 microns. The reflection probe had configurations with one fiber for illumination and three fibers for collection. Additionally, the entrance slit of the spectrometer was 50 μm, with a pixel size of 50 × 300 μm.

Sampling points with uniform texture were randomly selected, and spectra were acquired when the spectral curve became clear, stable, and exhibited no significant fluctuations. Each data point was averaged over three scans, and this process was repeated to collect 100 samples, with the mean calculated as the raw spectral data for each set of samples. The NIRQuest512 spectrometer is accompanied by SpectraSuite® software, which facilitates sampling, averaging, and exporting commands using scripts. All data were exported to an Excel file for storage and further analysis.

Subsequently, the 195 sets of samples with completed spectral data collection underwent fat content detection using the Soxhlet extraction method. The Soxhlet extraction method is widely recognized as the standard method for measuring fat content due to the principle that fat readily dissolves in organic solvents. After extracting the sample directly with anhydrous ether or petroleum ether, the solvent is evaporated, and the residue is dried to a constant weight, allowing for the calculation of the free fat content. The main steps include processing the pine nuts, packaging the pine nut powder, drying the samples, extracting the samples, weighing the extracted material, and calculating the results. It is important to note that the number of extraction cycles was set based on experiments with other nuts. Since pine nuts have a high fat content, to ensure data accuracy, the extraction cycle was set to 72 times as per the standard. The experiment was conducted in accordance with national food safety standards [29] (GB5009.6-2016), with fat content determination certified by the Heilongjiang Institute of Quality Supervision and Testing.

Subsequently, the 195 sets of samples with completed spectral data collection underwent fat content detection using the Soxhlet extraction method. The Soxhlet extraction method is widely recognized as the standard method for measuring fat content, involving several steps including slicing, packaging, drying, extraction, weighing, and result calculation. The experiment was conducted in accordance with national food safety standards [29] (GB5009.6-2016), with fat content determination certified by the Heilongjiang Institute of Quality Supervision and Testing.

### 2.3. Data Analysis

#### 2.3.1. Offline and Online Preprocessing of Spectral Data

MSC is a preprocessing algorithm designed to mitigate the scattering effects caused by surface properties of samples, such as variations in refractive index, particle size, and surface roughness. It is particularly suitable for diffuse reflectance spectroscopy due to its ability to effectively remove unwanted scattering effects from spectral data, thereby enhancing the accuracy of quantitative analysis. According to reference [30], MSC has demonstrated effective performance in processing the spectra of pine nuts compared to other preprocessing algorithms designed to mitigate scattering effects. For a specific spectrum, the MSC algorithm can be performed as follows.

First, the average spectrum $\bar{x}$ of the calibration set samples is calculated. Then, a linear regression operation, given by Equation (Equation 1), is conducted between each individual spectrum $x_{i}$ and the average spectrum $\bar{x}$.
(1)$$x_{i}=a_{i}\bar{x}+b_{i}$$

Calculate the slope $a_{i}$ and intercept $b_{i}$ based on the principle of least squares. Then, obtain $x_{i,MSC}$, which is given by Equation (Equation 2).
(2)$$x_{i,MSC}=\left(x_{i}-b_{i}\right)/a_{i}$$

During the research process, it was observed that, besides the inherent differences in physical and chemical properties between new and old pine nuts, the spectral data collection intervals varied significantly among different batches of samples. This resulted in distinct initial conditions for data collection in experiments. Experimental validation indicated that simply merging new and old data for scattering correction did not yield satisfactory results when predicting the behavior of the new sample set. In online model research, where the sample set is real-time and dynamic, it is imperative that new data undergo independent scattering correction from the original dataset. Additionally, since the preprocessed new data will iteratively update the parameters of the original predictive model, it is crucial that preprocessing does not deviate significantly from the original modeling dataset. The real-time and dynamic nature of new sample data necessitates online preprocessing. To address these challenges, we propose an enhanced version of the MSC algorithm, termed OMSC.

When the (N+1)-th new sample spectrum enters the preprocessing stage, the mean value of all (N+1) sample data is:(3)$${\bar{x}}_{n+1}=\frac{{\bar{x}}_{n}\times n+x_{n+1}}{n+1}.$$

According to the MSC principle, the data of the (N+1)-th sample obtained after MSC preprocessing is:(4)$$x_{n+1}=a_{n+1}{\bar{x}}_{n+1}+b_{n+1}.$$

At this point, $x_{n+1}$ has not been corrected and, by applying the least squares method to obtain $a_{n+1}$ and $b_{n+1}$, we have:(5)$$x_{n+1,MSC}=\frac{x_{n+1}-b_{n+1}}{a_{n+1}}.$$

**Theorem 1.** 
*Regret is the difference between the cumulative actual loss and the minimum loss under a fixed hypothesis known in advance. It can be represented as:*

$$R_{T}^{static\;}=\sum_{t=1}^{T}l_{t}\left(w_{t}\right)-\min_{w}\sum_{t=1}^{T}l_{t}\left(w\right).$$



In general [31], the regret bound is defined as the upper bound corresponding to the worst-case regret value of a certain online learning algorithm. If the regret bound of a certain online learning algorithm is a sub-linear function with respect to the number of iterations *T*, that is, $R_{T}=o\left(T\right)$, then this online learning algorithm can be considered ideal because, as *T* tends to infinity, the losses of the optimal offline algorithm and the online learning algorithm can be considered approximately equal. The proposed OMSC in this study does not affect the convergence of online learning algorithms, and this will be proven next.

**Theorem 2.** 
*Suppose the maximum deviation of the near-infrared spectral absorbance at the same wavelength between the samples is E, where E is a positive constant.*


**Theorem 3.** 
*If the upper bound of the regret value for the online learning algorithm in this study is R, then after preprocessing the newly added sample data with OMSC, the upper bound of the regret value for the final algorithm is R+O(E).*


**Proof.** To prove it by contradiction, try to assume that the statement is false; proceed from there, and at some point, you will arrive at a contradiction.
$${\bar{x}}_{n}=\frac{x_{1}+\cdots +x_{n}}{n}$$
$${\bar{x}}_{n+1}=\frac{x_{1}+\cdots +x_{n+1}}{n+1}$$
$${\bar{x}}_{n+1}-{\bar{x}}_{n}=\frac{\left(x_{1}-x_{n+1}\right)+\cdots +\left(x_{n}-x_{n+1}\right)}{n(n+1)}<\frac{E}{n+1}$$When the new sample data $x_{n+1}$ are preprocessed, it will cause a slight variation in $a_{n+1}$, and the maximum magnitude of the variation will not exceed $b_{n+1}$.
$$x_{n+1,MSC}=\frac{x_{n+1}-b_{n+1}}{a_{n+1}}$$According to the principle of the MSC algorithm, $x_{n,MSC}=\frac{x_{n}-b_{n}}{a_{n}}$, it can be inferred that:
$$\left|x_{n+1,MSC}-x_{n,MSC}\right|=O\left(\left|{\bar{x}}_{n+1}-{\bar{x}}_{n}\right|\right)<\frac{O(E)}{n+1}.$$Therefore, the loss function, $\Delta l_{t}\left(\cdot \right)=O\left(\left|x_{n+1,MSC}-x_{n,MSC}\right|\right)<\frac{O(E)}{n+1}$, and we can obtain:
$$R_{T,MSC}^{\mathrm{static}\;}=\sum_{t=1}^{T}l_{t}\left(w_{t}\right)+\Delta l_{t}\left(w_{t}\right)-\min_{w}\sum_{t=1}^{T}l_{t}\left(w\right)<R+O\left(E\right).$$   □

The pseudocode of the OMSC algorithm is represented in Algorithm 1.
**Algorithm 1:** The OMSC pseudo-code.Input: A set of NIR spectra collected for i samples $X_{i}$, ${\bar{x}}_{i}$ is reference NIR spectra of $X_{i}$, new NIR spectra $x_{i+1}$Output: MSC transformed spectra $x_{i+1,MSC}$, reference NIR spectra ${\bar{x}}_{i+1}$1:Compute the reference spectra of n+1 samples: ${\bar{x}}_{n+1}=\frac{{\bar{x}}_{n}\times n+x_{n+1}}{n+1}$2:residual: $r={\bar{x}}_{n+1}-\left(a_{n+1}\bar{x}+b_{n+1}\right)$3:Compute $a_{n+1}$ and $b_{n+1}$: $\min_{a_{n+1},b_{n+1}}∥{\bar{x}}_{n+1}-\left(a_{n+1}\bar{x}+b_{n+1}\right)∥$4:$x_{n+1,MSC}=\frac{x_{n+1}-b_{n+1}}{a_{n+1}}$5:**return** $x_{i+1,MSC}$

After the scattering correction, the spectral data are subjected to the S–G convolution smoothing process. This algorithm utilizes polynomial fitting and the least squares method to calculate the weighted average value of wavelength points within the window [32]. Its purpose is to eliminate the high-frequency noise carried by the original spectral data. The result of S–G algorithm preprocessing is not affected by other samples in the dataset. In both offline and online model studies, the data can be directly smoothed after scattering correction.

#### 2.3.2. Feature Extraction Methodology

The UVE algorithm can screen and remove the invalid information carried by the full spectrum data. It reduces the data size to within a reasonable limit and tries to ensure the amount of effective information as much as possible. The basic principle of UVE is to introduce a random noise matrix into the spectral data matrix and obtain the PLS regression model by cross-validation [33]. Due to the noise matrix and the original spectral data matrix having the same dimension, the regression coefficient matrix can be obtained, denoted as *B*. There exists a linear relationship between the spectral data matrix and the fat content matrix as follows:(6)Y=bX+e.

In the equation above, *b* is the regression coefficient vector, and *e* represents the error vector. The average and standard deviation of vector *b* in matrix *B* are divided to obtain *C*.
(7)$$C_{i}=\frac{\mathrm{mean}\left(b_{i}\right)}{std\left(b_{i}\right)}$$

Here, vector *i* represents the *i*-th column of the spectral data matrix, and mean (b) and std(b) represent the mean and standard deviation of vector *b*, respectively. By judging the absolute value of $C_{i}$, we consider whether to retain the *i*-th column vector in the spectral data matrix.

#### 2.3.3. Modeling Methodology

PLS is widely used in various fields for its stability. In particular, its excellent performance in dealing with multicollinearity makes it one of the most recognized regression algorithms in the field of NIRS analysis [34,35].

The recursive partial least squares (RPLS) regression algorithm is commonly used in regression analysis. RPLS updates the regression coefficients of the original model during the iterative process with newly added modeling data. In this way, it can extract information from the feature data of newly added pine nut samples [36,37].

RPLS involves operations with two important covariance matrices. The regression coefficients of the PLS model are calculated using matrix $X^{T}X\left(t\right)$, while the latent variables are obtained from matrix $X^{T}y\left(t\right)$. Here, *X* denotes the spectral feature matrix, and *y* represents the vector of actual fat content. When the feature data of the *t*-th sample in the new dataset are added to the sample database, $X^{T}X\left(t\right)$ and $X^{T}y\left(t\right)$ are recursively updated using Equations (Equation 8) and (Equation 9):(8)$$X^{T}X\left(t\right)=\lambda X^{T}X\left(t-1\right)+\hat{x}{\left(t\right)}^{T}\hat{x}\left(t\right),$$
(9)$$X^{T}y\left(t\right)=\lambda X^{T}y\left(t-1\right)+\hat{x}{\left(t\right)}^{T}\hat{y}\left(t\right),$$
where $X^{T}X\left(t\right)$ and $X^{T}y\left(t\right)$ respectively represent the replaced covariance matrices, and λ represents the forgetting factor (0<λ≤1), whose role is to facilitate the rate at which the original covariance matrix is updated through feedback. During the recursive calculation process, the *t*-th sample spectrum feature data x(t) and the true value data y(t) of the fat content both need to be standardized. The process of standardization involves an average vector and a standard deviation vector. The recursive updating process of the average and standard deviation vectors is shown in Equations (Equation 10) to (Equation 13):(10)$$\bar{x}\left(t\right)=\frac{N-1}{N}\bar{x}\left(t-1\right)+\frac{1}{N}x\left(t\right),$$
(11)$$\bar{y}\left(t\right)=\frac{N-1}{N}\bar{y}\left(t-1\right)+\frac{1}{N}y\left(t\right),$$
(12)$$\delta_{x}^{2}\left(t\right)=\frac{N-2}{N-1}\delta_{x}^{2}\left(t-1\right)+\frac{1}{N-1}{\left(x\left(t\right)-\bar{x}\left(t-1\right)\right)}^{2},$$
(13)$$\delta_{y}^{2}\left(t\right)=\frac{N-2}{N-1}\delta_{y}^{2}\left(t-1\right)+\frac{1}{N-1}{\left(y\left(t\right)-\bar{y}\left(t-1\right)\right)}^{2},$$
where *N* represents the number of all samples in the database at this time. The value of *N* varies according to the change in the number of iterations. Then, Equations (Equation 14) and (Equation 15) are used to calculate the standardized spectrum feature data $\hat{x}\left(t\right)$ and the true value data $\hat{y}\left(t\right)$ of fat content.
(14)$$\hat{x}\left(t\right)=\frac{x\left(t\right)-\bar{x}\left(t\right)}{\delta_{x}\left(t\right)}$$
(15)$$\hat{y}\left(t\right)=\frac{y\left(t\right)-\bar{y}\left(t\right)}{\delta_{y}\left(t\right)}$$

The initial series of values before recursive updating of the RPLS model can be calculated based on the feature data and the true values of fat content in the modeling dataset. These initial values are represented as $\bar{x}\left(0\right),\bar{y}\left(0\right),\delta_{x}\left(0\right),\delta_{y}\left(0\right),X^{T}X\left(0\right)$, and $X^{T}y\left(0\right)$.

#### 2.3.4. Fat Content Calibration Model

This study first establishes an offline PLS prediction model for the fat content of Korean pine nuts. The process of upgrading this offline model to an online learning model mainly revolves around the newly added sample dataset. The spectral data in the new dataset are preprocessed by the OMSC and S–G convolution smoothing algorithms, and the data then need to be reselected for features by UVE. With the offline model, the RPLS algorithm can be used to achieve online updating of the prediction model. The construction process of the offline and online learning models is shown in Figure 3.

#### 2.3.5. Model Validation

By adjusting and comparing parameters, the modeling process is fine-tuned, and the final model is evaluated and decided upon. In this study, the root mean square error (RMSE) and the coefficient of determination [38] $R^{2}$ are used as evaluation metrics for the regression model. Specifically, RMSE is divided into the root mean square error of cross-validation (RMSECV) and the root mean square error of prediction [39] (RMSEP) for the calibration set and prediction set, respectively. RMSE and $R^{2}$ are calculated using Equations (Equation 16) and (Equation 17), respectively:(16)$$RMSECV/RMSEP=\sqrt{\frac{\sum_{i=1}^{n}{\left(y_{i}-{\hat{y}}_{i}\right)}^{2}}{n}},$$
(17)$$R^{2}=\sqrt{\frac{\sum_{i=1}^{n}{\left({\hat{y}}_{i}-y_{i}\right)}^{2}}{\sum_{i=1}^{n}{\left({\hat{y}}_{i}-\bar{y}\right)}^{2}}},$$
where $y_{i}$ represents the actual measured values corresponding to the pine nut fat content in this study, ${\hat{y}}_{i}$ represents the predicted values, and $\bar{y}$ represents the average measured values.

## 3. Results

### 3.1. Sample Fat Content and Dataset Division

Before model construction, a reasonable dataset partition is necessary. The calibration set and the prediction set for testing the final performance of the model are divided in a ratio of 2:1. The composition of the dataset follows the principle of random sampling, and it is necessary to ensure that the actual values of the fat content in the calibration set have a larger range than those in the prediction set. The newly added pine nut samples in the calibration and prediction sets also need to follow the above two principles simultaneously. Since this study adopts cross-validation, the partition of the validation set is not shown here.

### 3.2. Establishment of the PLS Offline Model

#### 3.2.1. Preprocessing Results

The original spectrum of the pine nuts is shown in Figure 4a. The spectral data are greatly affected by surface particle scattering and artificial factors such as changes in light path. The variation in sample transmittance and absorbance is difficult to control within a small range, resulting in a dispersed data curve. Figure 4b shows the spectrum curve after pre-processing with the MSC and S–G algorithms in sequence. It can be seen that the pre-processed spectrum curve has a clearer profile. The dispersed phenomenon is eliminated to a great extent, and the absorption peaks are more obvious. The noise signals and clutter in the 900–1600 nm band are basically eliminated. The high-frequency noise at the 1500–1700 nm band is also somewhat reduced.

#### 3.2.2. Feature Selection Results

The UVE method was used to select characteristic spectral bands in the spectral data. The most stable part of the characteristics was selected to establish the calibration model. Figure 5a shows the stability output results of different bands after UVE selection. Experimental results showed that the best correlation between pine nut fat content and near-infrared spectral characteristic data was achieved when the top 70 most stable wavelength groups were selected. The purple coverage area in Figure 5b shows the 70 selected characteristic wavelength groups.

#### 3.2.3. Model Prediction Performance and Analysis

Table 1 shows the performance differences between the PLS prediction models based on the full spectrum and the selected features. The comparative analysis results are as follows: the PLS model based on the full spectrum has RMSECV and RMSEP of 7.9952 and 7.8754, respectively, and $R^{2}$ values of 0.9569 and 0.9177 for calibration and prediction sets, respectively. Compared with the UVE-PLS model with 70 selected features, the UVE-PLS model has better $R^{2}$ and RMSE. It can be seen that, under the premise of setting a reasonable number of selected features, the model performance is better when the input data undergo feature selection. In this study, after global optimization using grid search, UVE fixed the number of features in the input matrix of the model at 70, which reduces the data dimension as much as possible while carrying enough information to meet the modeling requirements.

Figure 6 shows the visualization of the correlation between the calibration and prediction sets. Based on the analysis of the graph, both the PLS and UVE-PLS models perform well during the calibration process, but there are significant differences in the correlation of the prediction sets. Due to the presence of ineffective information and interference, the output data points of the PLS model in Figure 6a are scattered and show weak correlation. Therefore, feature selection plays an important role in data preprocessing before modeling. The correlation between the optimized spectral data and the true value of the fat content is also improved.

It should be noted that the UVE-PLS model shows consistent correlation performance between the prediction and calibration sets, while the PLS model does not exhibit the same effect. This is because the original spectra of pine nuts contain a large amount of information and have a complex structure. The redundant information makes it easy for the model training process to overfit and cause significant deviation in the prediction set results. The UVE-PLS model improves the model performance from the perspective of optimizing the data quality. However, a model with superior performance still needs to withstand the test of different batches of prediction set samples.

### 3.3. Establishment of the RPLS Online Model

#### 3.3.1. Updated Feature Selection and Results

Before iteratively updating the covariance matrix of the original model with new sample spectra that have undergone online preprocessing, the new sample data must undergo feature selection. The difference is that the number of features selected by the UVE technique needs to be expanded at this stage. This adjustment is necessary because the introduction of new data alters the importance of the selected feature bands. The PLS model weight is determined by the eigenvalues of its covariance matrix.
(18)$$X^{T}X_{\mathrm{new}\;}=\Gamma^{T}\Lambda \Gamma$$
(19)$$\Lambda =\left[\begin{array}{cccc} \lambda_{1} & 0 & \cdots & 0\\ 0 & \ddots & \; & \vdots \\ \vdots & \; & \ddots & 0\\ 0 & \cdots & 0 & \lambda_{n} \end{array}\right]$$

In order to update the covariance matrix of the original model with the preprocessed spectral data of newly acquired samples, feature selection should also be performed on the new data. However, the number of features selected by UVE needs to be increased at this point because the weightings of the selected spectral bands may change with the inclusion of new data. In this study, we monitored the feature bands corresponding to the first 30 eigenvalues λ during the iterative process. Through experimentation, we found that the model performance was optimal when the number of features was expanded to 100. Figure 7a shows the 100 spectral features selected by UVE. Figure 7b–d show the feature selection results of the RPLS model after 0, 15, and 30 iterations, respectively. The blue regions represent the feature bands corresponding to the 30 largest updated eigenvalues. It can be observed from these figures that the selected features undergo dynamic changes during the iterative process, and the final selected features shown in red reflect the transfer of selected features for the spectral data of newly acquired samples.

It is important to note potential discrepancies in selected features between the new sample data and the original modeling data. This arises from differences in the feature selection range between the original sample dataset and the new dataset after the addition of new samples. The shift in feature wavelengths suggests that the initially selected features may not encompass some of the updated features in the dataset, necessitating an increase in the number of features.

#### 3.3.2. Comparison and Analysis of Prediction Performance

Note that the blue data points in Figure 8a,b represent the test results of the original MSC-PLS model and OMSC-RPLS model, respectively, without adjusting the number of input features (70 input features) for the new sample data. The blue data points in Figure 8c,d represent the test results of the MSC-RPLS model and OMSC-RPLS model, respectively, after increasing the number of input features (100 input features). The red points in Figure 8b,d represent the visualization output of the correlation results of the new data calibration (iteration). As can be seen from the figure, the data points preprocessed by OMSC are more evenly distributed, with only a few weak correlations. As the sample data are added one by one and after multiple iterations, the models also tend to be stable. The generalization ability of each model is mainly evaluated based on the test results of the prediction set. By observing the distribution of points, the predictive ability of the model can be preliminarily judged and analyzed.

It can be seen that, compared with the original MSC-PLS offline model, the updated RPLS online learning model generally has a stronger prediction ability for new samples. The reason for this phenomenon is that the offline model cannot accurately explore effective feature information contained in the new sample dataset. The red dots in Figure 8b,d are compactly and uniformly distributed, while in Figure 8c, the red dots are scattered. Due to the unsatisfactory preprocessing results of the new sample spectral data, the overall deviation of the updating and correction process is relatively large. Compared with traditional preprocessing methods, the OMSC algorithm makes the updating and correction process more in line with the requirements of improving model adaptability. The hollow blue circles reflect the direct relationship between the performance of the prediction set testing results and the strength of the model. It can be concluded that the testing results of the OMSC-RPLS model in Figure 8d are highly correlated and have better performance. The final conclusion needs to be determined through specific comparison analysis based on the RMSEP and other specific output parameters of each model in the testing set, as shown in Table 2.

Based on Table 2 and Figure 8, the predictive relationship and model performance regarding the fat content of the new pine nuts can be validated. The results indicate that the original offline model exhibits poor detection ability for new samples. Conversely, the online learning model established by the RPLS algorithm demonstrates a certain degree of generalization for new samples, aligning with the visualization effects depicted in Figure 8b–d. Through analysis of both the figure and the table, it is evident that both the MSC-RPLS model and the MSC-PLS model yield unsatisfactory testing results for the new sample prediction set at this juncture. This can be primarily attributed to the fact that, when the new sample data undergo MSC preprocessing, each new spectrum is subjected to linear regression analysis, with the average spectrum derived from the calibration sample set of the original offline model. The acquisition of the original reference spectrum remains independent of the additional data and remains unchanged, even with the inclusion of new data. Consequently, this preprocessing methodology only mitigates the disparity between the new sample spectrum data and the original calibration set of spectrum data, without considering the impact of new samples on the reference spectrum. As a result, a significant error occurs in the preprocessing stage of the new sample spectrum, thereby potentially compromising the final prediction outcomes of the model.

When the number of input features is increased from 70 in the original offline model to 100, the prediction accuracy of the model is higher. RMSEP decreased from 9.9039 to 3.5964, and $R_{p}^{2}$ increased from 0.8772 to 0.8945. The results in Figure 8 and Table 2 indicate that using OMSC to preprocess the original spectral data of new samples and increasing the number of features extracted by UVE can establish a prediction model with good detection ability for new pine nut samples in combination with the RPLS algorithm.

## 4. Discussion

NIRS is a rapid and effective technology for assessing the quality of agricultural products and food. Several studies have successfully correlated the nutritional content of nuts with spectral data [40,41,42]. However, a common challenge in practical application arises from the fact that the items being measured often arrive in batches, making it difficult to consistently match the physicochemical properties of the samples used during offline model development. Consequently, offline models may remain limited to feasibility studies and may fail to transition out of laboratory settings. This limitation stems from the requirement in offline learning that all training data be available during model development, with the model only becoming usable for predictions after training is complete. In contrast, online learning processes data sequentially, continuously updating the model (referred to as the offline model in this study) as real-time data become available. Nonetheless, this advantage of online learning introduces certain risks. Since the model processes one data point at a time and updates weights immediately after training, erroneous weight calculations resulting from faulty data can potentially lead the model astray. To mitigate this risk, this study thoroughly preprocesses new samples, ensuring alignment with the original reference spectra and thereby reducing the likelihood of online learning weight calculations veering off course from the source data, effectively minimizing residuals.

To validate the superiority of the online learning model, we focused on comparing the performance of three online models. We specifically discussed the impact of spectral data dimensionality on modeling work. Additionally, the sample quantity determines the dataset volume, thereby affecting optimization effectiveness. Having too many new samples would eliminate the advantage of iteratively updating model parameters instead of reestablishing the model. The number of online dataset samples should be kept within a reasonable range. In this study, data collection was conducted in batches, with a total of 75 sets of samples comprising the NIR_ONLINE dataset. The NIR_ONLINE dataset samples were organized into batches of 5, sequentially inputted into the online model for training, and real-time outputs of RMSEP and $R_{p}^{2}$ were obtained. This process was used to analyze the model quality mentioned in Table 3. Figure 9a,b depict the RMSEP and $R_{p}^{2}$ iteration curves for the online model. The results of the prediction set tests directly reflect the strength of the predictive model performance.

From Figure 9a,b, the OMSC-RPLS method can accurately extract effective feature information from the new sample dataset. Compared with traditional preprocessing methods, the OMSC and RPLS algorithms make the updating and correction process more consistent with the requirements of improving model adaptability. The OMSC-RPLS-100 model is initially unstable, and its accuracy is slightly lower than that of the OMSC-RPLS-70 model. However, as the model continues to iterate and the dataset is input in batches to the online model for training and prediction, new data gradually increase, and the OMSC-RPLS-70 model strictly controls the number of bands. When the number of feature bands gradually exceeds 70, the model accuracy gradually decreases. On the other hand, the accuracy of the OMSC-RPLS-100 model gradually increases, and when the number of samples in the NIR_ONLINE dataset reaches about 30, the model accuracy approaches the maximum value, and the weight tends to stop updating. The NIR_ONLINE dataset should ideally contain as few samples as possible. In this study, the sample size of the NIR_ONLINE dataset ranged from 10 to 60. The model validation results are shown in Table 3.

In general, a larger number of training samples leads to higher prediction accuracy. Following the design principle of minimizing the size of the online learning dataset, this study set the number of samples in the online partial correction set to 30. At this point, the enhanced prediction accuracy of the updated online learning model now exceeds that of the offline model, with both model RMSEP and $R_{p}^{2}$ approaching their maximum values. The prediction results meet the accuracy requirements. In future optimizations of the model through online learning, the proportion of online samples can serve as a reference. In similar detection tasks in the future, a more in-depth investigation can be conducted into the setting of the volume of the online learning dataset.

## 5. Conclusions

This study aims to address the limitations of conventional methods for determining the fat content of Korean pine nuts by proposing a comprehensive approach that leverages near-infrared spectroscopy (NIRS). Initially, a PLS offline prediction model was developed, which offers a rapid, nondestructive, and accurate detection method. However, recognizing the offline model’s shortcomings, such as limited generalization and inadequate sample preprocessing for model updates, we advanced an OMSC-RPLS online learning model. The OMSC algorithm performs independent scatter correction on new sample data without the need for the complete set of original model data, thereby enhancing prediction accuracy. Additionally, by expanding the feature selection range during preprocessing, the new model captures a greater proportion of relevant information, which, when fed into the RPLS model, leads to a progressively updated and stable regression model.

The results demonstrate that the online learning model significantly outperforms the original model in detecting new batches of samples, showcasing enhanced adaptability. This online model holds significant guiding and practical value for the determination of pine nut nutritional content, offering reference and application value for quantitative analysis, quality testing, and online learning research related to other nut varieties. Theoretically, the establishment of an updated sample database facilitates the long-term optimization and updating of models aimed at similar detection objectives.

Despite these advancements, the recursive PLS model is not the only option. Various incremental learning algorithms, such as online stochastic gradient descent, online AdaBoost, online SVM, and online k-means, remain underexplored. Furthermore, algorithms like online collaborative filtering, widely used on e-commerce platforms to update user preferences in real time, could also be applied to near-infrared spectroscopy. Compared to these emerging algorithms, recursive PLS offers higher interpretability of spectral data, and the predecessor algorithm, PLS, has been widely applied in near-infrared spectroscopy. After balancing the advantages of recursive PLS with those of emerging algorithms, this study chose the recursive PLS algorithm as the online model. Future research should investigate these alternatives to assess their applicability to NIRS.

Moreover, this study focused on near-infrared spectroscopy, while other potential nondestructive techniques, such as laser-induced breakdown spectroscopy (LIBS) and Raman spectroscopy, could also be applied to nondestructive food testing. Given LIBS’s high sensitivity and limited penetration, it could be a viable alternative for detecting the nutritional content of pine nuts. Future research should explore these techniques for potential applications in nondestructive analysis.

In conclusion, this study has delved deeply into chemometrics, machine learning, and online learning methods, seamlessly integrating them to establish a robust quality evaluation model for Korean pine nuts. This model not only characterizes the properties and determines the nutrient content of the nuts but also transitions from an offline to an online learning model, setting the stage for ongoing research and development in this field.

## Figures and Tables

**Figure 1 foods-13-02857-f001:**
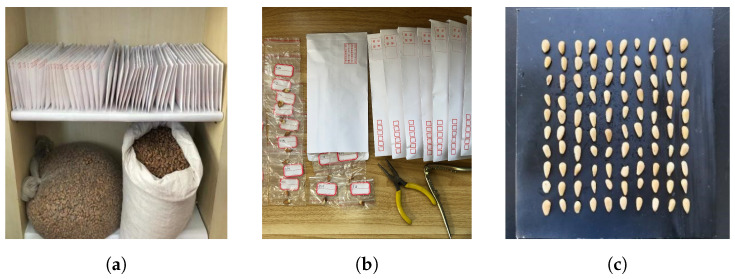
Samples to be tested in different states. (**a**) Part of the pine nut samples; (**b**) sample of pine nuts without shell; (**c**) sample of pine nut kernels without skin.

**Figure 2 foods-13-02857-f002:**
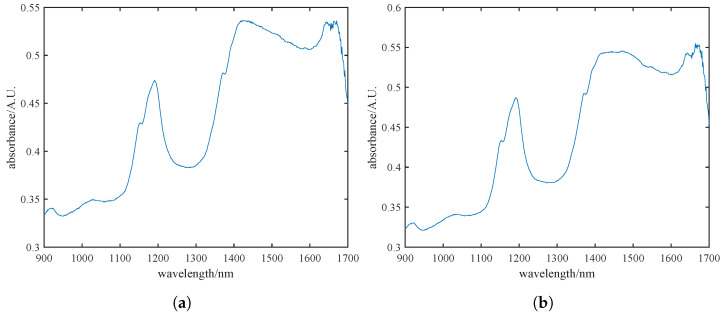
Near-infrared raw spectra of pine nut kernels with skins (**a**) and without skins (**b**), where the blue line in the figure represents the near-infrared spectral band of pine nuts.

**Figure 3 foods-13-02857-f003:**
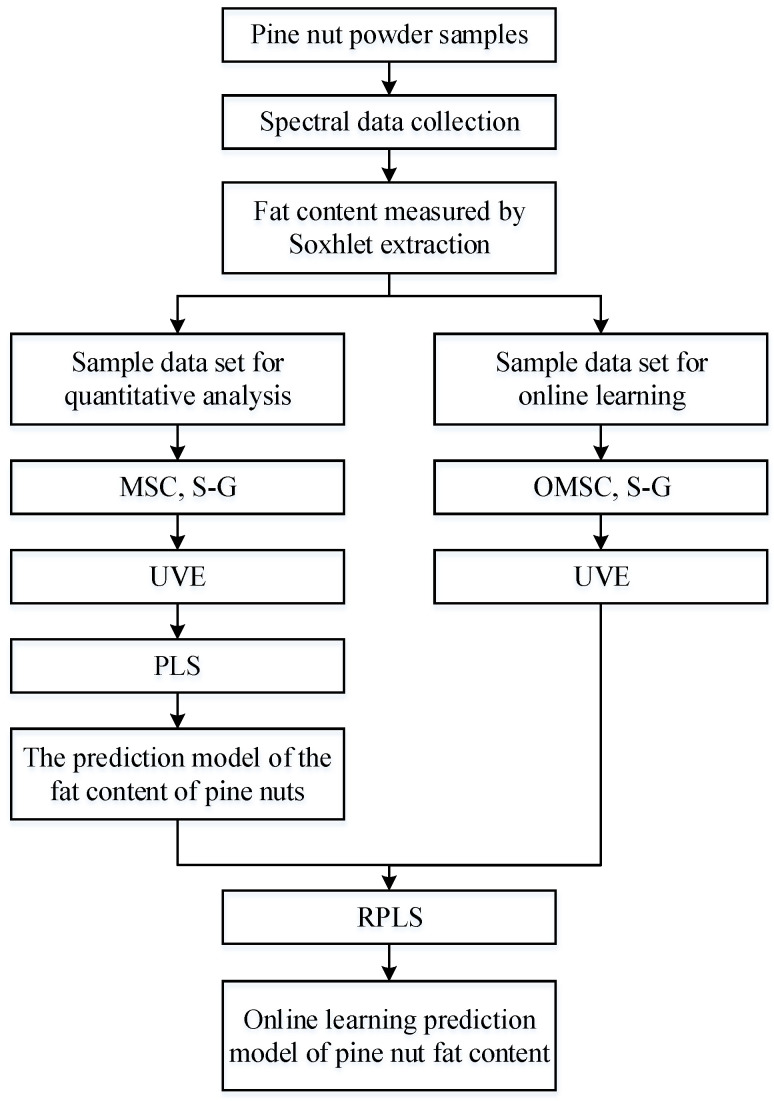
RPLS algorithm flow chart.

**Figure 4 foods-13-02857-f004:**
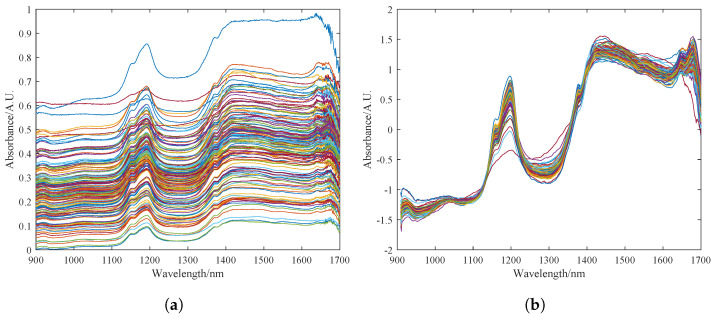
Preprocessing results of the sample dataset for fat content detection in Korean pine nuts. (**a**) Description of the raw spectrum of pine nuts. (**b**) Preprocessing results of MSC and S–G, where the colored lines in the figure represent the near-infrared spectral bands of different pine nuts.

**Figure 5 foods-13-02857-f005:**
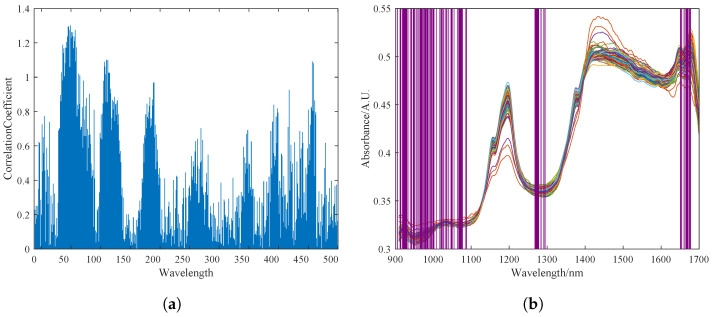
Feature selection of NIRS by UVE. (**a**) Stability diagram of spectral bands selected by UVE. (**b**) Output result of feature selection, where the blue lines of varying lengths in Figure (**a**) represent the importance of the near-infrared spectral bands as selected by the UVE algorithm, while the purple shading in Figure (**b**) indicates the spectral bands selected by the UVE algorithm.

**Figure 6 foods-13-02857-f006:**
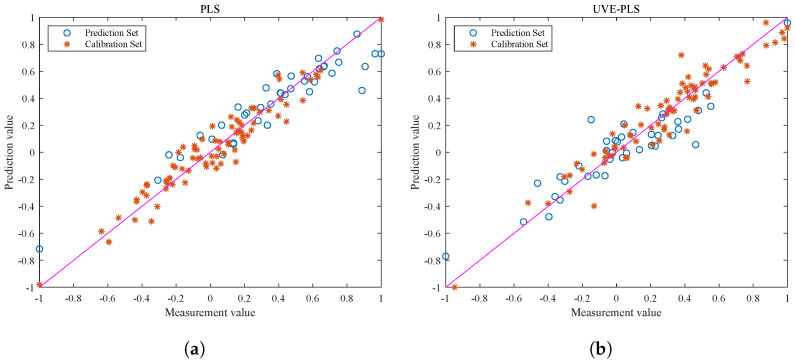
Visualizations of correlation of calibration set for fat content prediction of each correction model. (**a**) Visual output of PLS correlation. (**b**) Visual output of UVE-PLS correlation. The pink line serves as the reference baseline for the regression model. If the model’s output aligns with the true label data, the scatter points should fall on the pink line.

**Figure 7 foods-13-02857-f007:**
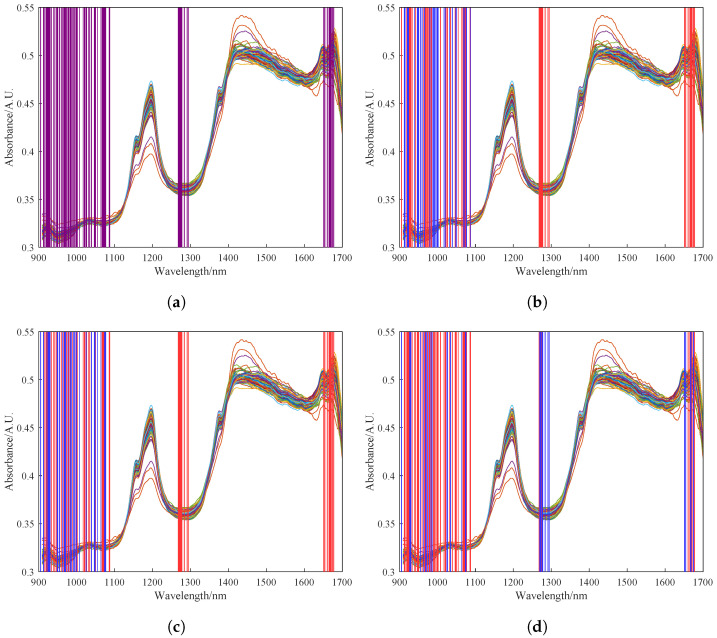
Feature results of feature wavelengths selected by UVE. (**a**) Feature results of feature wavelengths selected by UVE. (**b**) Feature selection results of 0 iterations of the RPLS model. (**c**) Feature selection results of 15 iterations of the RPLS model. (**d**) Feature selection results of 30 iterations of the RPLS model. The alternating red and blue shading in the figure represents the continual variation in the spectral bands selected by the UVE algorithm as the model iterates.

**Figure 8 foods-13-02857-f008:**
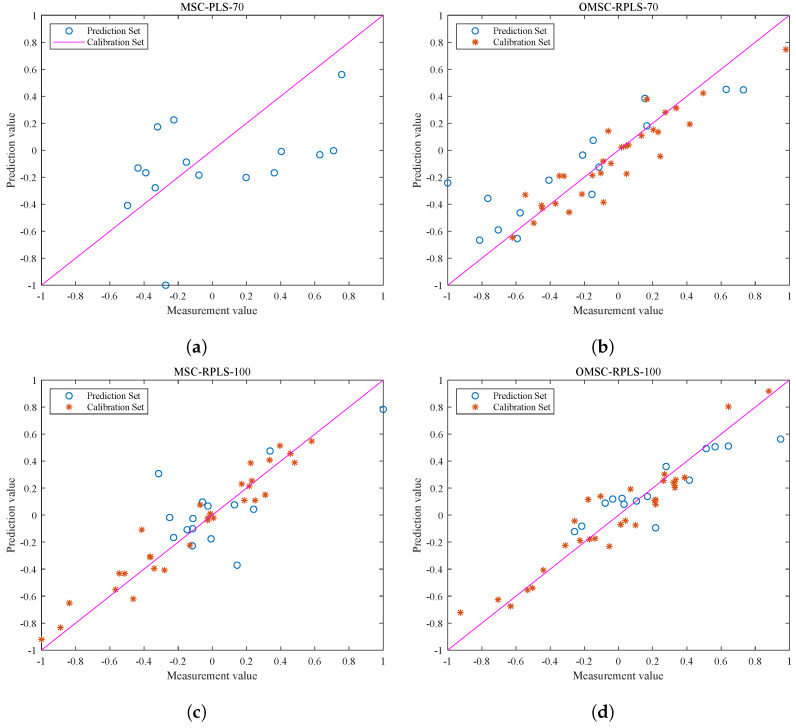
Correction and test results of each model. (**a**) Test results of new sample test data on the original PLS model. (**b**) Correction and test results of the OMSC-RPLS model (the number of features is 70). (**c**) Correction and test results of the MSC-RPLS model (the number of features is 100). (**d**) Correction and test results of the OMSC-RPLS model (the number of features is 100).

**Figure 9 foods-13-02857-f009:**
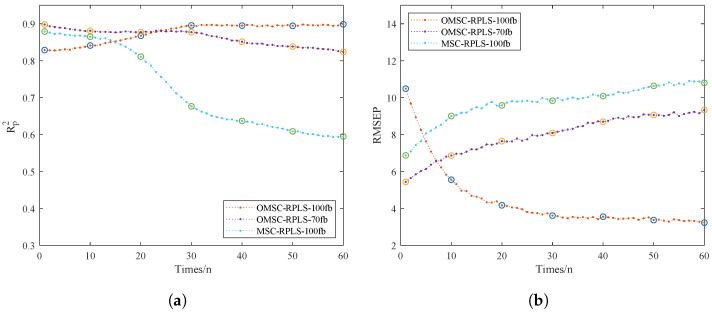
Iterative training results of the online models. (**a**) $R_{p}^{2}$. (**b**) RMSEP.

**Table 1 foods-13-02857-t001:** Parameter outputs of the calibration set for different modeling methods.

Model	Number of Features	Calibration (*n* = 80)		Prediction (*n* = 40)	
		RMSECV	$R_{c}^{2}$	RMSEP	$R_{p}^{2}$
PLS	511	7.9952	0.9569	7.8754	0.9177
UVE-PLS	70	6.4839	0.9590	7.3120	0.9464

$R_{c}^{2}$ represents the correlation coefficient for the calibration set, while $R_{p}^{2}$ denotes the correlation coefficient for the prediction set.

**Table 2 foods-13-02857-t002:** The outputs of the parameters of the predicted results of each model.

Model	Number of Input Features	Prediction (*n* = 15)		Calibration	
		RMSEP	$R_{p}^{2}$	RMSECV	$R_{c}^{2}$
MSC-PLS	70	24.6543	0.4525	–	–
OMSC-RPLS	70	9.9039	0.8772	5.1564	0.9236
MSC-RPLS	100	8.1060	0.6673	3.1320	0.9682
OMSC-RPLS	100	3.5964	0.8945	4.1832	0.9509

**Table 3 foods-13-02857-t003:** Comparison of the accuracy of the OMSC-RPLS-100 model with different numbers of datasets.

NIR_ONLINE	Prediction Set	
	**RMSEP**	$R_{p}^{2}$
10	5.5572	0.8410
20	4.1735	0.8676
30	3.6139	0.8954
40	3.5575	0.8951
50	3.3723	0.8945
60	3.2329	0.8986

## Data Availability

The original contributions presented in the study are included in the article, further inquiries can be directed to the corresponding author.

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
