# Peer review of "Towards Sustainable and Dynamic Modeling Analysis in Korean Pine Nuts: An Online Learning Approach with NIRS"

_foods, 2024, doi:10.3390/foods13172857_

Round 1

Reviewer 1 Report

Comments and Suggestions for Authors

The near-infrared band is a gold mine for non-destructive analysis, particularly when combined with multivariate data processing.

While both near-infrared technique and data processing algorithms are not new, it’s encouraging to see their application in a potential industrial process, such as the dynamic assessment of nut quality. This makes the paper worth reading and publishing, as it could inspire a broad audience and suggest applications in other areas.

Additionally, the focus on Korean nuts is original. These nuts can be considered functional food due to their beneficial fatty acid content. This is another valuable aspect of the paper. The clarity and effectiveness of Figures 4, 5, 6, 7, and 8 are commendable.

Comments and requests for minor revisions to improve the manuscript:

  1. Please specify the type of light source used in combination with the NIRQuest512 spectrometer.

  2. Clarify whether you used a fiber optic probe or a cavity for diffuse reflectance spectroscopy. If a fiber optic probe was used, please specify the number of fibers for illumination and detection, respectively.

  3. What is the slit size of the spectrometer? Is it fixed, or was it optimized for this study?

Lastly, I am unable to check for plagiarism, as I do not have access to the appropriate software.

Author Response

Date: 2024/9/3

Dear Reviewer,

Thank you for your constructive feedback on our manuscript titled “Towards Sustainable and Dynamic Modeling Analysis in Korean Pine Nuts: An Online Learning Approach with NIRS.” We have made the following revisions to address your comments:

Comment: Please specify the type of light source used in combination with the NIRQuest512 spectrometer.

Response: Thank you for your valuable suggestion. We have added that the light source used with the NIRQuest512 spectrometer was the HL-2000 Tungsten Halogen Light Sources (see manuscript, lines 156-164).

Comment: Clarify whether you used a fiber optic probe or a cavity for diffuse reflectance spectroscopy. If a fiber optic probe was used, please specify the number of fibers for illumination and detection, respectively.

Response: We appreciate your insightful comment. We have clarified that a fiber optic probe was used for diffuse reflectance spectroscopy. The fiber optics accessories included VIS-NIR fibers with core sizes of 200, 400, and 800 microns. The reflection probe had configurations with one fiber for illumination and three fibers for collection (see manuscript, lines 156-164).

Comment: What is the slit size of the spectrometer? Is it fixed, or was it optimized for this study?

Response: Thank you for your attention to detail. We specified that the entrance slit of the spectrometer is 50 µm and is adjustable. The pixel size is 50 x 300 µm (see manuscript, lines 156-164).

We appreciate your valuable input, which has helped enhance the clarity and precision of our manuscript. Thank you for considering our revised submission.

Yours faithfully,

Dr. Jiang
PhD Candidate
College of Computer Science, Northeast Forestry University
[email protected]

Reviewer 2 Report

Comments and Suggestions for Authors

In this study, the authors determined the fat content of Korean pine nuts using an NIR sensor, incorporating an online detection model based on RPLS. The authors can find my comments below.

·      Lines 29-31: The authors mention the benefits of Korean pine nuts. However, it is unclear whether these properties are specific to Korean pine nuts or to pine nuts in general. The sentence should clarify whether these benefits apply exclusively to Korean pine nuts.

·      Additional information about the Soxhlet methodology, used as the reference for fat content measurement, should be included in the manuscript. Please also specify the number of replications performed for the reference analysis.

·      The sample analysis should be described in more detail. For example, how were the spectra collected—was the skin removed? If so, have you compared the results with spectra collected with the skin on?

·      Including the study's limitations would be beneficial for potential readers.

·      Suggesting future research directions would provide valuable insights for studies in the field of spectral sensors for food quality analysis.

Comments on the Quality of English Language

The English language was fine, except for minor mistakes; therefore, I suggest the authors to read the manuscript one more time. 

Author Response

Date: 2024/9/3

Dear Reviewer,

Thank you for your valuable feedback on our manuscript titled ““Towards Sustainable and Dynamic Modeling Analysis in Korean Pine Nuts: An Online Learning Approach with NIRS.” We have addressed your comments as follows:

Comment: The authors mention the benefits of Korean pine nuts. However, it is unclear whether these properties are specific to Korean pine nuts or to pine nuts in general. The sentence should clarify whether these benefits apply exclusively to Korean pine nuts.

Response: Thank you for this important observation. In response to your suggestion, we have revised the manuscript to remove Korean pine nuts specifically. The discussion now focuses on pine nuts in general (see manuscript, lines 29-30).

Comment: Additional information about the Soxhlet methodology, used as the reference for fat content measurement, should be included in the manuscript. Please also specify the number of replications performed for the reference analysis.

Response: We appreciate your insightful comment. Additional information about the Soxhlet methodology, including the number of replications, has been added (see manuscript, lines 173-182). The Soxhlet extraction was performed 72 times in accordance with national standards.

Comment: The sample analysis should be described in more detail. For example, how were the spectra collected—was the skin removed? If so, have you compared the results with spectra collected with the skin on?

Response: Thank you for highlighting this aspect. Details on spectral data collection with and without pine nut skins have been provided (see manuscript, lines 133-140). We chose to use unshelled nuts with skins for detection.

Comment: Including the study's limitations would be beneficial for potential readers.

Response: We are grateful for your suggestion. Limitations and future research directions have been addressed (see manuscript, lines 523-538). Alternative algorithms and non-destructive techniques like LIBS are suggested for further investigation.

Comment: Suggesting future research directions would provide valuable insights for studies in the field of spectral sensors for food quality analysis.

Response: Thank you for your valuable input. Future research directions have been included alongside the study's limitations (see manuscript, lines 523-538). We propose exploring alternative algorithms and non-destructive techniques like LIBS for further investigation.

We believe these revisions have strengthened the manuscript and addressed your concerns. Thank you again for your insightful comments and for your consideration of our revised manuscript.

Yours faithfully,

Dr. Jiang
PhD Candidate
College of Computer Science, Northeast Forestry University
[email protected]